# Inferior Nutritional Status Significantly Differentiates Dialysis Patients with Type 1 and Type 2 Diabetes

**DOI:** 10.3390/nu15071549

**Published:** 2023-03-23

**Authors:** Anna Grzywacz, Arkadiusz Lubas, Stanisław Niemczyk

**Affiliations:** Department of Internal Medicine, Nephrology and Dialysis, Military Institute of Medicine–National Research Institute, Szaserów 128, 04-141 Warsaw, Poland

**Keywords:** diabetes, dialysis, hemodialysis, peritoneal dialysis, nutritional status, malnutrition, hospitalizations, all-cause death

## Abstract

Diabetes mellitus is currently the leading cause of end-stage renal disease. Assessing nutritional status is an important component of care in this group. This prospective observational study aimed to assess the nutritional status of type 1 and type 2 diabetes patients on hemodialysis or peritoneal dialysis and its relationship with hospitalizations and all-cause death. Adult patients with end-stage renal disease, treated with dialysis, and suffering from type 1 or type 2 diabetes, being treated with insulin, were included in the study. Exclusion criteria comprised other types of diabetes, the patient’s refusal to participate in the study, and severe disorders impacting verbal-logical communication. The nutritional status based on the Nutritional Risk Index, the Geriatric Nutritional Risk Index, fat distribution measures, and the Charlson Comorbidity Index was estimated for 95 Caucasian dialysis patients with type 1 (n = 25) or type 2 (n = 70) diabetes. Patients with type 1 diabetes exhibited significantly inferior nutritional status and increased nutritional risk than those with type 2 diabetes. Lower values of nutritional indices significantly differentiated patients with type 1 from those with type 2 diabetes, with ≥84% sensitivity and specificity. Inferior nutritional status was related to all-cause hospitalizations, whereas higher comorbidity was associated with a greater likelihood of cardiovascular hospitalizations and all-cause death. The significant difference between patients with type 1 and type 2 diabetes being treated with dialysis indicates that these patients should not be considered as a homogeneous group, while also considering the greater age of patients with type 2 diabetes.

## 1. Introduction

Diabetes mellitus (DM) is currently the leading cause of end-stage renal failure requiring renal replacement therapy [1,2,3]. Previous studies including dialysis patients with diabetes have focused on the division between the hemodialysis (HD) vs. peritoneal dialysis (PD) methods, without differentiating between the type of diabetes experienced by the patients [4,5,6,7,8,9,10,11]. The incidence of patients with type 1 and type 2 DM is sometimes provided in the characteristics of the study group, but differences between them have not been studied, probably due to the disproportion between the group size of patients with type 1 and type 2 diabetes [12,13,14]. Type 2 DM affects 90–95% of patients with diabetes in the general population and among patients requiring renal replacement therapy, while type 1 DM is less common [15,16]. Other types of diabetes are even less frequent [15]. Type 1 and type 2 DM are distinct diseases characterized by a different pathogenesis and clinical course [17], which from the nephrological perspective, seems to be overlooked when assessing dialysis patients.

High comorbidity worsens the prognosis of dialysis patients. The Charlson Comorbidity Index (CCI) has been validated to assess comorbidity and survival in dialysis patients [18,19,20]. In the dialysis population, patients with diabetes have a higher mortality rate compared to patients without diabetes [21]. Lower survival rates are also observed in patients with malnutrition, which is a common clinical problem in this group [22]. Dialysis patients with diabetes are characterized by significantly lower nutritional parameters and a higher incidence of malnutrition compared to patients without diabetes [23,24,25,26,27]. Among hemodialyzed patients, the Geriatric Nutrition Risk Index (GNRI) has been used to assess the nutritional status, as well as all-cause [4,5,28,29], cardiovascular [30], and infectious mortality [31]. The GNRI has also been used to assess nutritional status and prognosis in peritoneal dialysis patients [6,32], although it has not been proven to be a sufficiently sensitive indicator for screening malnutrition in this group [33]. As a screening tool for malnutrition in PD patients, the Nutrition Risk Index (NRI) can be used because it has high sensitivity, but low specificity compared to the Subjective Global Assessment (SGA) [34]. Its usefulness was tested in 283 patients, aged 12–65 years, on peritoneal dialysis, 53% of whom also had diabetes [34]. 

This study aims to assess the nutritional status of dialysis patients with type 1 and type 2 diabetes, using various accessible measures and indices, and to investigate the relationship between these nutritional parameters and comorbidity with hospitalizations and all-cause death.

## 2. Materials and Methods

This is a prospective observational study. Patients were recruited from the beginning of June 2018 to the end of April 2019. Laboratory tests and anthropometric measurements were performed at the time of inclusion in the study. Participants were followed for mortality and hospitalizations, from the time of inclusion in the study to the end of May 2020. The identification of patients’ comorbidity was based on the analysis of the medical records from the dialysis center. 

### 2.1. Ethics Approval and Consent to Participate

The study was conducted in accordance with the Declaration of Helsinki and was approved by the Bioethics Committee of the Military Institute of Medicine (protocol code 29/WIM/2017 of 9 August 2017). Informed consent was obtained from all subjects involved in the study.

### 2.2. Eligibility Criteria

Inclusion criteria were age ≥ 18 years, end-stage renal disease (ESRD) with the duration of hemodialysis or peritoneal dialysis ≥ 3 months, type 1 or type 2 diabetes lasting ≥ 3 months, and the use of insulin. 

The diagnosis of type 1 or type 2 diabetes was based on the analysis of medical records and medical history.

Exclusion criteria comprised other types of diabetes, the patient’s refusal to participate in the study, and severe disorders impacting verbal-logical communication.

### 2.3. Study Measurements

Waist and hip circumference, weight, and height were measured before the hemodialysis, or at the control visit in peritoneal dialysis patients. Usual weight was assessed from the medical documentation. 

The waist to hip ratio (WHR) was calculated according to the formula: WHR = waist circumference (cm)/hip circumference (cm).

Body mass index (BMI) was calculated using the formula: BMI = body weight (kg)/height (m)^2^.

NRI was calculated according to the formula [34]: NRI = [1.519 × albumin (g/L)] + (41.7 × current weight/usual weight).

NRI distinguishes four levels of nutrition-related risk: major (NRI < 83.5), moderate (NRI 83.5–97.5), mild (NRI 97.5–100), and no risk (NRI > 100) [34].

The GNRI formula was as follows [35,36]:GNRI = [1.489 × albumin (g/L)] + (41.7 × current weight/ideal body weight).

Ideal body weight (IBW) was calculated according to the formula [29,36,37]:IBW = 22 kg/m^2^ × height (m)^2^.

GNRI also defines four levels of nutrition-related risk: major (GNRI < 82), moderate (GNRI 82–92), low (GNRI 92–98), and no risk (GNRI > 98) [35].

Laboratory tests were performed in the Department of Laboratory Diagnostics at the Military Institute of Medicine-National Research Institute. Blood samples were obtained, along with routine periodic laboratory tests, at the beginning of the hemodialysis, or during control visits in peritoneal dialysis patients. 

Serum total protein (laboratory norm (n): 6.4–8.3 g/dL) and albumin (n: 3.9–4.9 g/dL) were obtained using a colorimetric method; total cholesterol (n: 120–200 mg/dL), low density lipoprotein (LDL, n: 50–130 mg/dL), high density lipoprotein (HDL, n for female: 35–65 mg/dL, for male: 35–55 mg/dL), and triglycerides (n: 35–165 mg/dL) were assessed by enzymatic colorimetric reactions; glycated hemoglobin (HbA1c) was assessed by turbidimetric inhibition immunoassay (n: 4.8–5.9%); and serum C-reactive protein (CRP, n: 0.0–0.8 mg/dL) was measured by the immunoturbidimetric method using a Cobas c 501 analyzer (Roche Diagnostics). Blood morphology was obtained using the automated hematology analyzer SYSMEX XN-1000 (white blood cell (WBC) n: 4.0–10.0 × 10^9^/L, hemoglobin (HGB) n: 11.0–18.0 g/dL). 

Fasting glucose was measured by the patient at home with a glucometer for 7 consecutive days after laboratory testing.

Comorbidity was assessed according to CCI [38]. The causes of hospitalizations and the date of death were taken from hospital discharge summary reports kept by the dialysis center archives. 

The dialysis was considered adequate if the Kt/V ≥ 1.2 and the urea reduction ratio (URR) ≥ 65% for HD or Kt/V ≥ 1.7 for PD criteria were met [39,40].

### 2.4. Statistical Analysis

The data were presented as the mean with standard deviation, or as a number with occurring frequency. The normal distribution of variables was checked with the Shapiro–Wilk test. Differences between two variables with normal distribution were estimated using the *t*-test; otherwise, with the Mann–Whitney test, and the between categorical variables were assessed with the Chi-squared test. The Kruskal–Wallis analysis of variance was used to evaluate the differences in groups of three or more variables. The point-biserial correlation was performed for one dichotomous variable, or the Spearman test was conducted for association analysis. Hospitalizations were considered as continuous (sum of hospitalization incidents) or dichotomous (was, or was not) variables. Univariate logistic regression analysis was used to investigate the association with the type of diabetes. The receiver operating characteristic (ROC) curve was plotted, with the cut-off value determined using the Youden index method, to show the discriminatory properties of the nutritional indices. Missing data were deleted pairwise from the analysis. The statistical test result was significant if the two-tailed *p*-value was <0.05. Statistical software STATISTICA v 12.0 (StatSoft Inc., Cracow, Poland) was used for statistical analysis.

## 3. Results

A total of 95 people were included in the study (38% women and 62% men; age 63.2 ± 14.1 years). All participants were Caucasian, which reflects the high homogeneity of the investigated population. Type 1 diabetes was diagnosed in 25 (26.3%) participants, and type 2 diabetes was found in 70 (73.7%) of them. 

### 3.1. Dialysis Method

A total of 71 (74.7%) patients were treated with hemodialysis and 24 (25.3%) with peritoneal dialysis. The characteristics of hemodialysis and peritoneal dialysis patients, grouped according to the type of diabetes experienced, are presented in Table 1.

In both the HD and PD groups, patients with type 1 DM, compared to patients with type 2 DM, were significantly younger, had a longer history of diabetes with a higher HbA1c, and exhibited a lower body weight, BMI, waist circumference, hip circumference, and WHR. They also had substantially lower NRI and GNRI scores, as well as lower comorbidity. Hemodialyzed patients with type 1 DM also had higher HDL and mean fasting glucose. The CRP concentration in the study groups slightly exceeded the upper reference limit (Table 1).

Based on performed analyses, peritoneal dialysis patients were younger than those undergoing hemodialysis (58.2 ± 12.5 vs. 64.8 ± 14.3; *p* = 0.018) and had significantly lower comorbidity (6.9 ± 1.6 vs. 8.4 ± 2.3 points in CCI; *p* = 0.005 with 6.9 ± 13.0 vs. 4.2 ± 13.8% of estimated 10-year survival; *p* = 0.026). Likewise, the serum albumin concentration was lower in PD vs. HD individuals (3.5 ± 0.3 vs. 3.8 ± 0.4; *p* < 0.001). Total cholesterol and LDL were much higher in people on PD vs. those on HD (179.9 ± 34.5 vs. 148.1 ± 43.1; *p* < 0.001 and 110.4 ± 27.7 vs. 85.7 ± 36.5; *p* < 0.001, respectively). Moreover, more patients treated with PD had adequate dialysis compared to patients treated with HD (81.3% vs. 47.1%; *p* = 0.024). Nevertheless, the HD and PD groups did not differ significantly regarding diabetes care, anthropometric parameters, WHR, NRI, and GNRI. For this reason, these parameters were further compared between patients with type 1 and type 2 diabetes only.

Most of the study participants, 20 (80.0%) patients with type 1 DM and 48 (68.6%) patients with type 2 DM, were under outpatient diabetes care, with no significant differences between the groups (*p* = 0.277). The average frequency of visits to a diabetologist in the year preceding the study was about 2 (Table 1).

The adopted criterion for the adequacy of dialysis was met by 73.7% of patients with type 1 DM and by 47.8% of patients with type 2 DM (*p* = 0.067).

### 3.2. Nutritional Status

Table 2 shows the distribution of particular grades of nutrition and nutrition-related risk in different indices in the groups divided according to the type of diabetes. 

Patients with type 1 DM had significantly inferior nutritional status compared to patients with type 2 diabetes. No nutritional risk was observed in 48.0% of patients with type 1 diabetes and 94.3% of patients with type 2 diabetes, according to the NRI. However, according to the GNRI, 60.0% of patients with type 1 DM and 95.8% with type 2 DM exhibited no nutritional risks.

The waist circumference ≤ 80 cm in women and ≤94 cm in men applied to 52.0% of patients with type 1 diabetes and to only 2.9% of those with type 2 diabetes. Waist circumference ≤ 88 cm in women and ≤102 cm in men applied to 72.0% of patients with type 1 diabetes and 17.1% with type 2 diabetes. The recommended WHR values for Europeans (<0.85 for women and <0.90 for men) were met in 32.0% of patients with type 1 DM and 2.9% with type 2 DM. 

To estimate the potential relationships, a univariable logistic regression analysis was performed, revealing that the investigated nutritional measures and comorbidity were significantly associated with the type of diabetes (Table 3).

The ROC analysis showed good and similar discriminatory properties of the considered leading nutritional indices in the differentiation of patients with type 1 or type 2 diabetes (Table 4, Figure 1, Figure 2 and Figure 3). 

### 3.3. Hospitalizations and Mortality

A total of 95 patients were followed by 15.6 ± 5.1 months from the time of first assessment. During this period, 22 people died, with no significant differences between the study groups. Two participants were lost from observation: a man with type 1 diabetes and a woman with type 2 diabetes, both hemodialyzed. We could not determine the number of hospitalizations of these two patients, but it was established that they were alive at least until the end of May 2020. Table 5 contains the frequency of cardiovascular, infectious, and all-cause hospitalizations, with the all-cause patient deaths divided according to the type of diabetes. 

Table 6 shows the results of the point-biserial correlation of all-cause death and hospitalizations as dichotomic variables. Only the number of points obtained according to the Charlson Comorbidity Index significantly correlated with the occurrence of all-cause death (r = 0.235; *p* = 0.022). All-cause hospitalizations were significantly positively associated with CRP concentration (r = 0.214; *p* = 0.039) and negatively with waist circumference (r = −0.250; *p* = 0.016) and hip circumference (r = −0.232; *p* = 0.026). The NRI and GNRI scores were negatively correlated with all-cause hospitalizations (r = −0.192; *p* = 0.066 and r = −0.192; *p* = 0.065, respectively). The occurrence of infectious hospitalizations was significantly related only to serum CRP concentration (r = 0.209; *p* = 0.044).

Table 7 contains the results of the quantity associations regarding all-cause, cardiovascular, and infectious hospitalizations. In this approach, the number of all-cause hospitalizations significantly negatively correlated with the number of points in the NRI (r = −0.211; *p* = 0.043) and the GNRI (r = −0.210; *p* = 0.044) and waist circumference (r = −0.256; *p* = 0.013), and positively correlated with total cholesterol concentration (r = 0.221; *p* = 0.034).

The number of cardiovascular hospitalizations correlated positively with the number of points in CCI, (r = 0.227; *p* = 0.028) and negatively with the CCI% estimated 10-year survival rate (r = −0.215; *p* = 0.038).

## 4. Discussion

In our study, all the nutritional parameters of patients with type 1 diabetes were significantly lower than those of participants with type 2 diabetes (Table 1 and Table 2). Among patients with type 1 DM, the prevalence of severe malnutrition was 8.0%, and mild to moderate malnutrition ranged from 32.0% to 44.0%, depending on the evaluation method (NRI or GNRI), while in type 2 DM, it was 0.0%, and from 4.2% to 5.7%, respectively. 

Malnutrition is a disorder caused by poor diet, impaired digestion, or disturbed nutrient utilization. Protein-energy wasting (PEW) is a recommended term to describe states of undernutrition with decreased nutrient intake and/or increased catabolism in maintenance dialysis patients. PEW is the state of reduced stores of protein and energy fuels in the body [41]. The most serious form of PEW is cachexia. According to another definition, loss of muscle is necessary to recognize cachexia. However, loss of adipose tissue is possible, but not required for diagnosis. The criteria for distinguishing between PEW and cachexia have been proposed, but often, the distinction between these conditions is blurred, and sometimes the nomenclature is used interchangeably [42]. There are four main categories in the diagnosis of PEW: biochemical criteria (cholesterol, serum albumin, and prealbumin), body mass (total body fat, unintentional weight loss, and BMI), measures of muscle mass (urinary creatinine appearance, mid-arm muscle circumference, and total muscle mass), and dietary protein and energy intake [41,43]. PEW can be diagnosed when at least three out of the abovementioned four diagnostic categories are abnormal. Optimally, each of them should be documented at least three times, at 2- to 4-week intervals. Other elements, such as appetite, food intake, energy expenditure, body composition, laboratory tests—including inflammation markers—various measures of body composition, and multiple nutritional scoring systems—such as SGA, dialysis malnutrition score, or Malnutrition Inflammation Score (MIS)—should also be considered [41]. 

MIS was validated as a screening tool for malnutrition in hemodialysis [44] and is also valuable for peritoneal dialysis patients [45,46]. It is a subjective tool requiring experienced staff to complete the questionnaire properly [29,47]. In everyday clinical practice, such a detailed assessment of all patients in terms of PEW would be problematic.

A simple and objective tool for assessing the nutritional status and associated risk in this group is the GNRI, which was derived from the NRI [36]. The NRI was developed by Buzby et al. to identify malnutrition or the risk associated with this status and related complications in young adult surgical patients [48]. Because of problems in establishing the usual weight in the elderly population, Bouillanne replaced “usual weight” with “ideal body weight”, creating the Geriatric Nutritional Risk Index [35,36]. In their study, the Lorentz formula was used to calculate IBW [35,36,49]. An alternative method used in this study to assess the GNRI is to calculate the ideal body weight corresponding to a BMI of 22 kg/m^2^ [29,36,37]. Initially, the GNRI was not an index of malnutrition, but a nutrition-related risk index [35]. However, in later studies among dialysis patients, the GNRI was used as an indicator of both malnutrition and prognosis [4,5,6,28,29,30,31,32]. In our work, the NRI and GNRI scores negatively correlated with all-cause hospitalizations (r = −0.192; *p* = 0.066 and r = −0.192; *p* = 0.065, respectively).

The results of our study show that BMI is not an appropriate screening method for detecting malnutrition among patients with diabetes on dialysis treatment when adopting population norms. According to BMI values, only one person was underweight, a woman with type 1 diabetes treated with peritoneal dialysis. In the presented study, most patients with type 1 diabetes had a BMI within the normal range, while most patients with type 2 diabetes met the criteria for first-class obesity. On the other hand, BMI values for PEW in chronic kidney disease (<23 kg/m^2^ [42]) included 48.0% of patients with type 1 DM and only 5.7% of patients with type 2 DM, and the difference was significant.

In various studies, BMI showed prognostic value in patients with ESRD. A higher BMI was connected with a lower risk of death, while a lower BMI was associated with a worse prognosis [50,51]. The curve illustrating the relationship between BMI and survival is J-shaped, with BMI in the range of 30–34.9 kg/m^2^ being the most favorable [52].

An important parameter is also the distribution of adipose tissue in the body. The assessment of metabolically unfavorable central obesity can be made by measuring the waist to hip ratio (WHR). According to the WHO recommendation for Europeans, a waist circumference > 80 cm in women and >94 cm in men is associated with an increased risk of metabolic complications, and a waist circumference > 102 cm or WHR ≥ 0.90 in men and a waist circumference > 88 cm or WHR ≥ 0.85 in women significantly increased the risk [53]. 

In our study, most patients with type 1 diabetes met the recommended values for waist circumference (52% and 72%, depending on the threshold). However, the gender target of WHR was achieved by only 32% of participants with type 1 DM. In patients with type 2 DM, these values were met by only a few patients. There are studies proving that WHR compared to BMI is a stronger predictor of all-cause death and death from cardiovascular causes in the general population [54,55]. In large epidemiological studies, WHR was also a predictor of coronary artery disease and death [54,56]. There are very little data on whether similar relationships will also occur in the population of patients with end-stage renal disease. Postorino et al. conducted a prospective cohort study in a group of 537 patients with ESRD, finding that in BMI-adjusted Cox models, waist circumference was a direct predictor of all-cause and cardiovascular mortality (*p* < 0.001), while BMI was inversely related (*p* < 0.001) to these results. The incidence of all-cause and cardiovascular death was greater in patients with a lower BMI (below the median) and a larger waist circumference (at least the median), while the incidence of cardiovascular and all-cause death was lower in patients with a higher BMI (at least the median) and a smaller waist circumference (below the median). The study proves that abdominal obesity is a significant risk factor for overall and cardiovascular mortality in patients with ESRD [55]. In the conducted study, waist circumference and hip circumference negatively correlated with the incidence of all-cause hospitalizations (r = −0.250; *p* = 0.016 and r = −0.232; *p* = 0.026, respectively), and the number of all-cause hospitalizations significantly negatively correlated with waist circumference (r = −0.256; *p* = 0.013). The NRI and GNRI indices were also negatively associated with all-cause hospitalizations, which could be the result of inferior nutritional status. 

In a Polish study involving 61 dialysis patients with type 1 or type 2 DM, the independent risk factors for all-cause death among diabetic patients treated with peritoneal dialysis were hypoalbuminemia and older age, while in hemodialyzed patients, they were hypoalbuminemia and low serum cholesterol [11]. In our study, however, serum total cholesterol positively correlated with all-cause hospitalizations (r = 0.221; *p* = 0.034).

Cardiovascular diseases are the leading cause of death among dialysis patients, including those with diabetes. Inflammation is also responsible for an increased risk of death from cardiovascular disease. C-reactive protein is the most commonly used marker of inflammation. In the population of patients with ESRD, CRP is an active factor in the atherosclerotic phenomenon [57]. A study conducted on 163 hemodialysis patients showed that patients with elevated CRP (≥1 mg/dL) had significantly lower 5-year survival rates and a substantially higher risk of death [58]. However, in our study, the groups did not differ significantly in the mean CRP concentration, which was slightly above the reference limit (Table 1). The occurrence of all-cause hospitalizations, as well as infectious hospitalizations, were significantly associated with serum CRP concentration (r = 0.214; *p* = 0.039 and r = 0.209; *p* = 0.044, respectively). The meta-analysis of 19 cohort studies, including 10,739 hemodialyzed patients, revealed that the GNRI had a significant negative association with all-cause mortality, cardiovascular events, and cardiovascular mortality [59]. We did not find associations between markers of nutritional status and mortality during our observation. In our study, only the number of points obtained according to CCI, reflecting greater comorbidity, significantly correlated with the all-cause death occurrence (r = 0.235; *p* = 0.022). In the observational study conducted on a similar group of 100 dialyzed patients with type 1 or type 2 DM, 23 patients died during 16.0 ± 5.0 months of observation. The deceased group also had greater comorbidity, according to the CCI, compared to the surviving participants (*p* = 0.013), and the history of stroke or transient ischemic attack were independent risk factors of all-cause death (HR 3.15, ±95% confidence interval (CI): 1.34–7.39; *p* = 0.009), while regular physical activity reduced that risk (HR 0.26, ±95% CI: 0.08–0.87; *p* = 0.029) [60].

Moreover, the number of cardiovascular hospitalizations also correlated positively with the number of points according to the CCI (r = 0.227; *p* = 0.028) and negatively correlated with the CCI% estimated 10-year survival rate (r = −0.215; *p* = 0.038). The lack of differences in mortality and hospitalizations between the groups of patients with type 1 and type 2 DM, despite the significantly older age and higher comorbidity in the group of patients with type 2 diabetes, may result from the compensation of the risk associated with better nutritional status. In addition, patients with type 1 DM compared to patients with type 2 DM had significantly higher mean fasting glycemia and HbA1c, despite comparable diabetes care. 

The conducted analyses showed numerous significant differences between patients with type 1 and type 2 DM, keeping in mind differences in age regarding estimated anthropometric parameters, nutritional status, and nutrition-related risk. Moreover, we showed that common nutritional indices, such as BMI, WHR, and GNRI, differ significantly between type 1 and type 2 DM dialysis patients, statistically enabling differentiation between these diseases. This means that patients with type 1 and type 2 diabetes should not be considered as a homogeneous group of “patients with diabetes” in research work and clinical practice. Although the proportions of type 1 DM and type 2 DM patients in the dialysis population are similar to those of PD and HD patients, the differences between dialysis methods are well-studied. Nevertheless, further research on a larger group is needed to confirm our findings concerning differences between diabetes types in the dialysis population.

The presented study has some limitations. The first is a relatively small number of included patients resulting from the eligibility criteria. The second is the unequal size of the investigated subgroups, reflecting a lower prevalence of type 1 diabetes in the dialysis population. In order to partially compensate for this disproportion, patients with type 1 DM were recruited from three additional dialysis centers in central Poland. Third, the follow-up time was very short because of the framework of the project. Additionally, there was a limited number of examined parameters regarding nutritional status. However, the full assessment of PEW was not the aim of this study. The exclusion of isolated cases of patients with severe verbal-logical communication disturbances, who were unable to provide information about diabetes care or medical history, in our opinion, did not affect the sample’s representativeness. The last issue is that the study was conducted only in the Caucasian population, which may limit the extrapolation of conclusions to dialysis patients of other races who also suffer from diabetes. Despite these limitations, we believe the results of the conducted analysis are reliable and relevant. 

## 5. Conclusions

Among Caucasian patients with type 1 or type 2 diabetes mellitus who are being treated with dialysis, inferior nutritional status and increased nutritional risk were found among patients with type 1 diabetes. Lower values of nutritional indices significantly differentiated patients with type 1 from patients with type 2 diabetes. The NRI and GNRI are valuable tools to assess nutritional status and nutritional risk, as well as to predict all-cause hospitalizations, in this patient population. The numerous significant differences between patients with type 1 and type 2 diabetes undergoing dialysis therapy indicate that these patients should not be considered as a homogeneous group. Nonetheless, type 1 diabetes in otherwise younger patients is responsible for a greater metabolic and nutritional derailment than that observed in much older subjects with type 2 diabetes.

## Figures and Tables

**Figure 1 nutrients-15-01549-f001:**
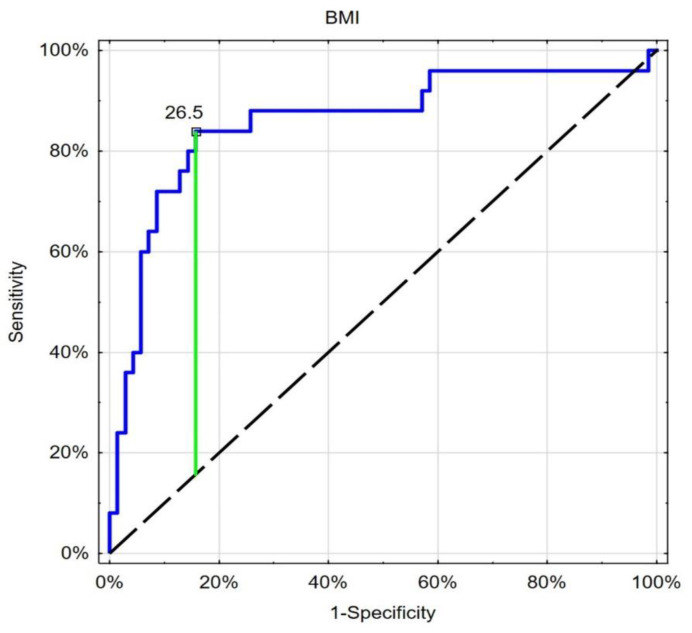
ROC analysis (blue line) of BMI in the differentiation between type 1 or type 2 diabetes, with the best cut-off point based on the Youden index (green line). Black dashed-reference line.

**Figure 2 nutrients-15-01549-f002:**
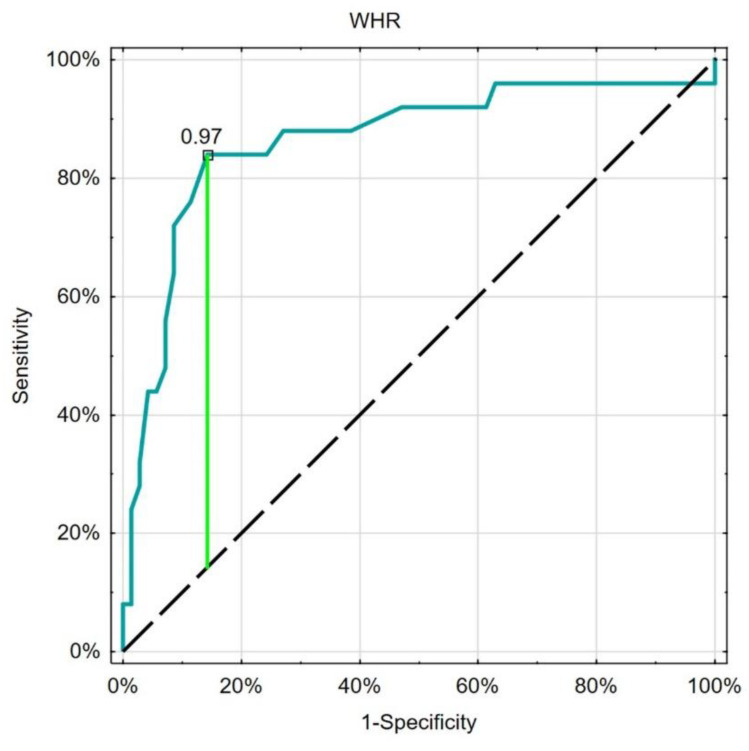
ROC analysis (turquoise line) of WHR in the differentiation between type 1 or type 2 diabetes, with the best cut-off point based on the Youden index (green line). Black dashed-reference line.

**Figure 3 nutrients-15-01549-f003:**
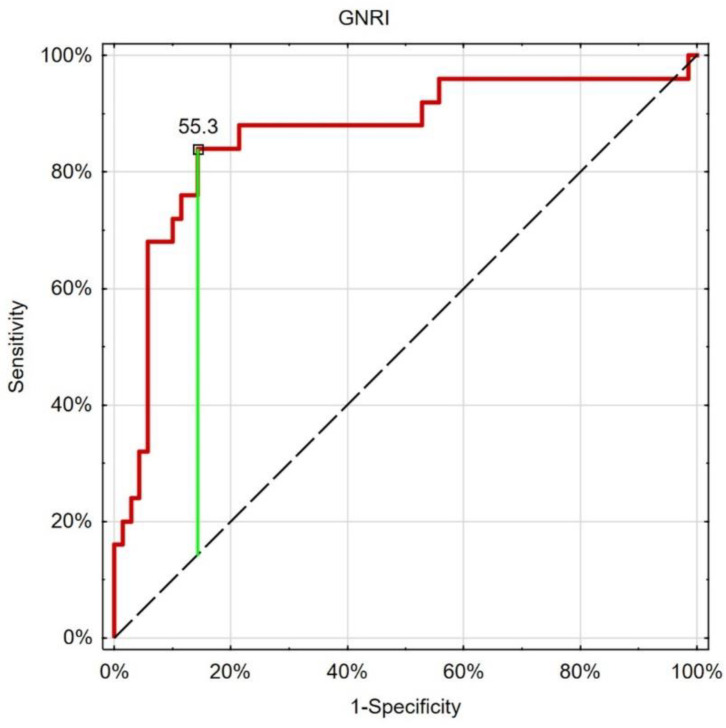
ROC analysis (red line) of GNRI in the differentiation between type 1 or type 2 diabetes, with the best cut-off point based on the Youden index (green line). Black dashed-reference line.

**Table 1 nutrients-15-01549-t001:** Characteristics of the hemodialysis and peritoneal dialysis patients, grouped according to the type of diabetes.

	Hemodialysis N = 71 M 43 (60.6%) F 28 (39.4%)			Peritoneal dialysis N = 24 M 16 (66.7%) F 8 (33.3%)		
	Type 1 DM N = 17 M 9 (52.9%) F 8 (47.1%)	Type 2 DM N = 54 M 34 (63.0%) F 20 (37.0%)	*p*	Type 1 DM N= 8 M 5 (62.5%) F 3 (37.5%)	Type 2 DM N = 16 M 11 (68.7%) F 5 (31.3%)	*p*
	**Mean ± SD**	**Mean ± SD**		**Mean ± SD**	**Mean ± SD**	
Age (years)	**46.7 ± 13.3**	**70.5 ± 8.9**	**<0.001**	**44.8 ± 10.2**	**64.9 ± 6.9**	**<0.001**
Duration of diabetes (years)	**33.1 ± 10.6**	**20.8 ± 9.9**	**<0.001**	**29.4 ± 10.5**	**17.4 ± 8.1**	**0.005**
Duration of dialysis (months)	38.7 ± 36.5	34.2 ± 26.9	0.811	**57.0 ± 101.9**	**30.6 ± 30.9**	0.878
Weight (kg)	**71.4 ± 18.3**	**89.9 ± 22.4**	**<0.001**	**71.0 ± 16.6**	**91.2 ± 14.2**	**0.005**
Height (cm)	168.7 ± 8.0	169.3 ± 8.7	0.816	171.6 ± 13.4	168.4 ± 10.7	0.524
BMI (kg/m^2^)	**25.3 ± 8.0**	**31.2 ± 6.4**	**<0.001**	**23.8 ± 3.0**	**32.3 ± 5.0**	**<0.001**
Waist circumference (cm)	**91.0 ± 16.5**	**111.6 ± 15.0**	**<0.001**	**89.1 ± 10.3**	**111.6 ± 10.6**	**<0.001**
Hip circumference (cm)	**98.3 ± 19.1**	**106.4 ± 14.0**	**0.002**	**95.0 ± 7.2**	**103.7 ± 8.0**	**0.017**
WHR	**0.93 ± 0.14**	**1.05 ± 0.1**	**<0.001**	**0.94 ± 0.1**	**1.08 ± 0.1**	**<0.001**
Total cholesterol (mg/dL)	153.0 ± 38.7	146.5 ± 44.6	0.479	185.9 ± 37.7	176.9 ± 33.6	0.558
Triglycerides (mg/dL)	125.4 ± 91.9	167.35 ± 102.8	0.068	123.3 ± 56.5	181.6 ± 122.8	0.375
HDL (mg/dL)	**56.3 ± 20.4**	**40.6 ± 11.3**	**0.001**	44.6 ± 21.3	47.4 ± 18.9	0.284
LDL (mg/dL)	83.7 ± 32.1	86.3 ± 38.1	0.957	124.4 ± 26.3	103.4 ± 26.4	0.080
Total protein (g/dL)	6.6 ± 0.8	6.6 ± 0.6	0.968	6.5 ± 0.9	6.5 ± 0.5	0.832
Albumin (g/dL)	3.8 ± 0.6	3.9 ± 0.4	0.319	3.3 ± 0.4	3.5 ± 0.3	0.198
Mean fasting glucose * (mg/dL)	**157.6 ± 41.2**	**133.1 ± 27.9**	**0.008**	144.8 ± 23.2	130.8 ± 27.9	0.076
HbA1c (%)	**7.5 ± 1.3**	**6.6 ± 1.4**	**0.010**	**8.2 ± 1.8**	**7.0 ± 1.0**	**0.047**
Number of visits to a diabetologist in the last year	2.5 ± 2.2	2.1 ± 2.2	0.576	1.6 ± 2.0	2.0 ± 1.7	0.593
CRP (mg/dL)	1.0 ± 2.0	1.2 ± 2.4	0.067	1.4 ± 1.7	1.0 ± 0.8	0.462
WBC (×10⁹/L)	7.0 ± 1.3	7.2 ± 2.5	0.500	7.9 ± 3.0	7.5 ± 1.3	0.708
HGB (g/dL)	10.8 ± 1.4	10.8 ± 1.4	0.963	11.2 ± 1.1	11.4 ± 1.5	0.815
CCI (points)	**6.5 ± 2.6**	**9.0 ± 1.9**	**<0.001**	**5.5 ± 1.2**	**7.6 ± 1.3**	**<0.001**
CCI estimated 10-year survival (%)	**17.3 ± 24.3**	**0.1 ± 0.5**	**<0.001**	**17.6 ± 17.3**	**1.6 ± 5.2**	**0.003**
IBW	62.8 ± 5.9	63.2 ± 6.4	0.803	65.2 ± 10.1	62.6 ± 7.9	0.503
NRI	**105.4 ± 14.9**	**117.6 ± 13.8**	**<0.001**	**95.8 ± 8.0**	**114.7 ± 9.3**	**<0.001**
GNRI	**104.2 ± 14.9**	**116.4 ± 13.7**	**<0.001**	**94.8 ± 8.0**	**113.6 ± 9.3**	**<0.001**

* measured with a glucometer by the patient at home for 7 consecutive days after laboratory testing. Bolded values are statistically significant. Abbreviations: N—number, F—female, M—male, HD—hemodialysis, PD—peritoneal dialysis, BMI—body mass index, WHR—waist to hip ratio, HDL—high density lipoprotein, LDL—low density lipoprotein, HbA1c—glycated hemoglobin, CRP—C-reactive protein, WBC—white blood cell, HGB—hemoglobin, CCI—Charlson Comorbidity Index, IBW—ideal body weight, NRI—nutritional risk index, GNRI—geriatric nutritional risk index.

**Table 2 nutrients-15-01549-t002:** The distribution of particular grades of nutrition and nutrition-related risk in different indices in the groups divided according to the type of diabetes.

	Type 1 Diabetes N = 25 (26.3%) F 11 (44.0%) M 14 (56.0%) HD 17 (68.0%) PD 8 (32.0%)	Type 2 Diabetes N = 70 (73.7%) F 25 (35.7%) M 45 (64.3%) HD 54 (77.1%) PD 16 (22.9%)	*p*
NRI < 83.5 (major nutrition-related risk)	**2 (8.0%)**	**0 (0.0%)**	**<0.001**
NRI 83.5–97.5 (moderate nutrition-related risk)	**6 (24.0%)**	**3 (4.3%)**	
NRI 97.5–100 (low nutrition-related risk)	**5 (20.0%)**	**1 (1.4%)**	
NRI > 100 (no nutrition-related risk)	**12 (48.0%)**	**66 (94.3%)**	
GNRI < 82 (major nutrition-related risk)	**2 (8.0%)**	**0 (0.0%)**	**<0.001**
GNRI 82–92 (moderate nutrition-related risk)	**1 (4.0%)**	**1 (1.4%)**	
GNRI 92–98 (low nutrition-related risk)	**7 (28.0%)**	**2 (2.8%)**	
GNRI > 98 (no nutrition-related risk)	**15 (60.0%)**	**67 (95.8%)**	
Waist circumference F ≤ 80 cm M ≤ 94 cm	**13 (52.0%)**	**2 (2.9%)**	**<0.001**
Waist circumference F > 80 cm M > 94 cm	**12 (48.0%)**	**68 (97.1%)**	
Waist circumference F ≤ 88 cm M ≤ 102 cm	**18 (72.0%)**	**12 (17.1%)**	**<0.001**
Waist circumference F > 88 cm M > 102 cm	**7 (28.0%)**	**58 (82.9%)**	
WHR F < 0.85 M < 0.90	**8 (32.0%)**	**2 (2.9%)**	**<0.001**
WHR F ≥ 0.85 M ≥ 0.90	**17 (68.0%)**	**68 (97.1%)**	
BMI < 18.5 kg/m^2^ underweight	**1 (4.0%)**	**0 (0.0%)**	**<0.001**
BMI 18.5–24.9 kg/m^2^ normal weight	**17 (68.0%)**	**7 (10.0%)**	
BMI 25–29.9 kg/m^2^ overweight	**4 (16.0%)**	**24 (34.92%)**	
BMI 30–34.9 kg/m^2^ obesity class I	**2 (8.0%)**	**26 (37.1%)**	
BMI 35–39.9 kg/m^2^ obesity class II	**0 (0.0%)**	**8 (11.4%)**	
BMI > 40 kg/m^2^ obesity class III	**1 (4.0%)**	**5 (7.1%)**	
BMI < 23 kg/m^2^	**12 (48.0%)**	**4 (5.7%)**	**<0.001**

Bolded values are statistically significant. Abbreviations: N—number, F—female, M—male, NRI—nutritional risk index, GNRI—geriatric nutritional risk index, WHR—waist to hip ratio, BMI—body mass index.

**Table 3 nutrients-15-01549-t003:** Results of univariable logistic regression analysis, with the type of diabetes as the dependent variable.

	Odds Ratio *	95% Confidence Interval	*p*
BMI (kg/m^2^)	0.769	0.675–0.877	<0.001
Waist circumference (cm)	0.883	0.837–0.931	<0.001
Hip circumference (cm)	0.930	0.881–0.981	0.008
WHR	0.235 × 10^−6^	0.206 × 10^−10^–0.264 × 10^−3^	<0.001
NRI	0.896	0.847–0.947	<0.001
GNRI	0.867	0.808–0.930	<0.001
CCI (points)	0.462	0.322–0.664	<0.001
CCI estimated 10-year survival (%)	1.198	1.057–1.358	0.005

* the odds ratio according to change from type 1 to type 2 diabetes. Abbreviations: BMI—body mass index, WHR—waist to hip ratio, NRI—geriatric nutritional risk index, GNRI—geriatric nutritional risk index, CCI—Charlson Comorbidity Index.

**Table 4 nutrients-15-01549-t004:** Results of ROC analysis of the leading nutritional indices.

	Cut-Off	Sensitivity	Specificity	AUC	*p*
BMI	26.5	0.840	0.843	0.858	<0.001
WHR	0.97	0.840	0.857	0.863	<0.001
GNRI	55.3	0.840	0.857	0.863	<0.001

Abbreviations: ROC—receiver operating characteristics, BMI—body mass index, WHR—waist to hip ratio, GNRI—geriatric nutritional risk index, AUC—area under the curve.

**Table 5 nutrients-15-01549-t005:** The frequency of cardiovascular, infectious, and all-cause hospitalizations, with all-cause patient deaths divided according to the type of diabetes.

	Type 1 Diabetes	Type 2 Diabetes	*p*
	N = 24	N = 69	
All-cause hospitalizations	15 (62.5%)	35 (50.7%)	0.319
Cardiovascular hospitalizations	5 (20.8%)	24 (34.8%)	0.204
Infectious hospitalizations	7 (29.2%)	14 (20.1%)	0.370
	N = 25	N = 70	
All-cause deaths	5 (20.0%)	17 (24.1%)	0.663

Abbreviation: N—number.

**Table 6 nutrients-15-01549-t006:** Point-biserial correlations of all-cause death and hospitalizations.

	All-Cause Hospitalizations		Cardiovascular Hospitalizations		Infectious Hospitalizations		All-Cause Deaths	
	r	*p*	r	*p*	r	*p*	r	*p*
BMI (kg/m^2^)	0.097	0.355	−0.045	0.669	−0.018	0.864	−0.067	0.522
Waist circumference (cm)	**−0.250**	**0.016**	−0.084	0.422	−0.026	0.802	−0.067	0.519
Hip circumference (cm)	**−0.232**	**0.026**	−0.150	0.150	−0.011	0.913	−0.126	0.224
WHR	−0.116	0.270	0.046	0.664	−0.046	0.661	0.042	0.685
Total cholesterol (mg/dL)	0.150	0.152	0.073	0.488	0.134	0.200	−0.170	0.099
Albumin (g/dL)	−0.073	0.490	−0.021	0.844	0.004	0.969	−0.078	0.452
CRP (mg/dL)	**0.214**	**0.039**	0.091	0.386	**0.209**	**0.044**	0.031	0.766
NRI	−0.192	0.066	−0.049	0.639	−0.014	0.893	0.096	0.357
GNRI	−0.192	0.065	−0.049	0.639	−0.014	0.892	0.095	0.359
CCI (points)	−0.003	0.978	0.203	0.051	−0.125	0.233	**0.235**	**0.022**
CCI estimated 10-year survival (%)	0.053	0.616	−0.190	0.068	0.054	0.605	−0.196	0.057

Bolded values are statistically significant. Abbreviations: BMI—body mass index, WHR—waist to hip ratio, CRP—C-reactive protein, NRI—nutritional risk index, GNRI—geriatric nutritional risk index, CCI—Charlson Comorbidity Index.

**Table 7 nutrients-15-01549-t007:** Quantity correlation analysis of all-cause, cardiovascular, and infectious hospitalization numbers.

	All-Cause Hospitalizations		Cardiovascular Hospitalizations		Infectious Hospitalizations	
	r	*p*	r	*p*	r	*p*
BMI (kg/m^2^)	−0.133	0.205	0.007	0.951	0.060	0.565
Waist circumference (cm)	**−0.256**	**0.013**	−0.080	0.446	−0.003	0.980
Hip circumference (cm)	−0.195	0.061	−0.133	0.206	0.003	0.976
WHR	−0.133	0.202	0.054	0.611	−0.002	0.988
Total cholesterol (mg/dL)	**0.221**	**0.034**	0.129	0.219	0.173	0.097
Albumin (g/dL)	−0.153	0.142	−0.032	0.759	−0.056	0.592
CRP (mg/dL)	0.147	0.159	0.109	0.297	0.154	0.140
NRI	**−0.211**	**0.043**	−0.010	0.929	−0.001	0.990
GNRI	**−0.210**	**0.044**	0.009	0.932	0.002	0.987
CCI (points)	−0.050	0.632	**0.227**	**0.028**	−0.118	0.259
CCI estimated 10-year survival (%)	0.089	0.398	**−0.215**	**0.038**	−0.118	0.259

Bolded values are statistically significant. Abbreviations: BMI—body mass index, WHR—waist to hip ratio, CRP—C-reactive protein, NRI—nutritional risk index, GNRI—geriatric nutritional risk index, CCI—Charlson Comorbidity Index.

## Data Availability

Not applicable.

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
