# Peer review of "Inferior Nutritional Status Significantly Differentiates Dialysis Patients with Type 1 and Type 2 Diabetes"

_nutrients, 2023, doi:10.3390/nu15071549_

Round 1

Reviewer 1 Report (New Reviewer)

This is a very important study on differentiating individuals with diabetes who are also CKD patients on dialysis. I very much appreciate that the authors underlined how patients with Type I and Type II DM should not be considered as one homogeneous group. The manuscript is well written and the study is well-conceptualized. I have a few comments, please see below, that in my opinion would improve the present manuscript:

1. Could you please revise the statement lines 51-55: "The study aimed to assess the nutritional status in patients with type 1 and type 2 diabetes on dialysis treatment using different accessible measures and indices and to investigate the relationship between these nutritional parameters and comorbidity, all- cause hospitalizations, cardiovascular, and infectious hospitalizations, as well as all-cause death".

Perhaps reword to be a bit more clear as to what the aim of the study is: This study aims to assess the nutritional status in patients with type 1 and type 2 DM and its impact on dialysis using various accessible measures and indices and to investigate the relationship between these nutritional parameters and comorbidity, all-cause hospitalizations and all-cause mortality.

2. How were patients with previous kidney transplant/one kidney viewed? Inclusion or exclusion criteria?

3. The statement that  HD and PD groups did not differ regarding anthropometric parameters, WHR, NRI, GNRI and diabetes care (Line 154) should be substantiated. For example, people on PD had HGB that was an entire point higher compared to people on HD. Likewise, NRI and GNRI were lower in PD vs. HD individuals. Total cholesterol was much higher in people on PD vs on HD. Could you please justify the appropriateness of conducting downstream analyses on only two groups: Type 1 and Type 2 DM without accounting for the dialysis modality.

4. Could you please provide the rationale for using the univariate logistic model rather than multivariate? It could be prudent to use a fully adjusted model, since the two groups were different for predictors such as age, obesity etc.  Additionally, it is unclear what the dependent variable was: was it type 1 or type 2 DM? I understand that you may have reasons why you chose to do it this way, if you could please provide some more rationale it would improve the clarity of the analysis choice.

5.Sentence on line 357-360 doesn't make sense, seems to be missing a word or a couple of words

Author Response

Much respected Reviewer,

Thank you for your review and constructive feedback on the structure of our article. While revising the manuscript, we considered all suggestions and comments. We believe that the manuscript has significantly improved in value after these corrections. We hope that it meets the expectations of the Reviewer as it stands and will be considered worthy of publication in the Nutrients.

Best regards,

Authors.

This is a very important study on differentiating individuals with diabetes who are also CKD patients on dialysis. I very much appreciate that the authors underlined how patients with Type I and Type II DM should not be considered as one homogeneous group. The manuscript is well written and the study is well-conceptualized. I have a few comments, please see below, that in my opinion would improve the present manuscript:

  1. Could you please revise the statement lines 51-55: "The study aimed to assess the nutritional status in patients with type 1 and type 2 diabetes on dialysis treatment using different accessible measures and indices and to investigate the relationship between these nutritional parameters and comorbidity, all- cause hospitalizations, cardiovascular, and infectious hospitalizations, as well as all-cause death".

Perhaps reword to be a bit more clear as to what the aim of the study is: This study aims to assess the nutritional status in patients with type 1 and type 2 DM and its impact on dialysis using various accessible measures and indices and to investigate the relationship between these nutritional parameters and comorbidity, all-cause hospitalizations and all-cause mortality.

Thank you for the tips on improving the style of the article. We have modified the aim of the study, including your suggestions (lines 59-62): “This study aims to assess the nutritional status in dialysis patients with type 1 and type 2 diabetes using various accessible measures and indices and to investigate the relationship between these nutritional parameters and comorbidity with hospitalizations and all-cause death”.

  1. How were patients with previous kidney transplant/one kidney viewed? Inclusion or exclusion criteria?

Much respected Reviewer, thank you for this remark. We did not exclude from the study patients with a previous kidney transplant or one kidney. However, these were isolated cases.

  1. The statement that HD and PD groups did not differ regarding anthropometric parameters, WHR, NRI, GNRI and diabetes care (Line 154) should be substantiated. For example, people on PD had HGB that was an entire point higher compared to people on HD. Likewise, NRI and GNRI were lower in PD vs. HD individuals. Total cholesterol was much higher in people on PD vs on HD. Could you please justify the appropriateness of conducting downstream analyses on only two groups: Type 1 and Type 2 DM without accounting for the dialysis modality.

Much respected Reviewer, we have divided the sentence showing the differences between the PD and HD groups into shorter ones, which makes the text more understandable (lines 157-167):

“Based on performed analyses, peritoneal dialysis patients were younger than those undergoing hemodialysis (58.2 ±12.5 vs. 64.8 ±14.3; p = 0.018) and had significantly smaller comorbidity (6.9 ±1.6 vs. 8.4 ±2.3 points in CCI; p = 0.005 with 6.9 ±13.0 vs. 4.2 ±13.8 % of estimated 10-year survival; p = 0.026). Likewise, serum albumin concentration was lower in PD vs. HD individuals (3.5 ±0.3 vs. 3.8 ±0.4; p < 0.001). Total cholesterol and LDL were much higher in people on PD vs. on HD (179.9 ±34.5 vs. 148.1 ±43.1; p < 0.001 and 110.4 ±27.7 vs. 85.7 ±36.5; p < 0.001, respectively). Moreover, more patients treated with PD had adequate dialysis compared to HD patients (81.3% vs. 47.1%; p = 0.024). Nevertheless, the HD and PD groups did not differ significantly regarding diabetes care, anthropometric parameters, WHR, NRI, and GNRI. For this reason, these parameters were further compared between patients with type 1 and type 2 diabetes only.”

  1. Could you please provide the rationale for using the univariate logistic model rather than multivariate? It could be prudent to use a fully adjusted model, since the two groups were different for predictors such as age, obesity etc.  Additionally, it is unclear what the dependent variable was: was it type 1 or type 2 DM? I understand that you may have reasons why you chose to do it this way, if you could please provide some more rationale it would improve the clarity of the analysis choice.

Thank you for this comment. We didn’t mind finding out the best predictor for the type of diabetes recognition because it is not proper from the etiological point of view. Thus, multivariable logistic regression analysis was not performed. However, we wanted to show the non-linear association between CCI and nutrition indexes and different diabetes types. Moreover, the prediction values of investigated variables were confirmed in further ROC analysis. We corrected the results section (lines 193-195): “To estimate the potential relations, a univariable logistic regression analysis was performed, and showed that the investigated nutritional measures and comorbidity were significantly associated with the type of diabetes (Table 3).”

Moreover, we mentioned that “..univariable logistic regression analysis was performed and showed that the investigated nutritional measures and comorbidity were significantly associated with the type of diabetes.” Thus, the dependent variable was defined as “type of diabetes”. However, according to the comment, we stated this precisely in table 3 title: “Results of univariable logistic regression analysis with the type of diabetes as the dependent variable”.

5.Sentence on line 357-360 doesn't make sense, seems to be missing a word or a couple of words.

Much respected Reviewer, we have enriched the fragment concerning the cited work with additional information (lines 369-375).

“In the observational study conducted on a similar group of 100 dialyzed patients with type 1 or type 2 DM, during 16.0 ± 5.0 months of observation, 23 patients died. The deceased group also had greater comorbidity according to CCI compared to the surviving participants (p = 0.013), and the history of stroke or transient ischemic attack were independent risk factors of all-cause death (HR 3.15 ±95 % confidence interval (CI): 1.34 - 7.39; p = 0.009), while regular physical activity reduced that risk (HR 0.26, ±95 % CI: 0.08 - 0.87; p = 0.029) [60].”

We hope that the corrected passage is now understandable.

Reviewer 2 Report (New Reviewer)

This study compared patients with Type 1 and Type 2 diabetes undergoing hemodialysis or peritoneal dialysis for nutritional status and clinical outcomes. This research was the first to compare the results of hemodialysis or peritoneal dialysis in Type 1 and Type 2 diabetes. However, there are some issues to address.

#1 The first time an abbreviation is used in the manuscript, it should be defined. In that regard, grammatical mistakes were common throughout and should be revised. Similarly, there were a few instances where commas were used instead of periods to mark the decimal place when writing a number. Periods should be used in standard American English.

#2 Why was 22 kg/m2 used for the ideal body weight formula? Ideal body weight differs with sex and age. Since the groups are made up of various sexes and ages, does using 22 kg/m2 skew the results the GNRI?

#3 Since the NRI was developed for young adults, how well does it measure the nutritional status of an older adult population?

#4 Since the study only includes Caucasians, would the lack of other races limit the generalization of the results? If so, this should be addressed in the limitations section. 

Author Response

Much respected Reviewer,

Thank you for your review and constructive feedback on the structure of our article. While revising the manuscript, we tried to take into account all suggestions and comments. We believe that the manuscript has significantly improved in value after these corrections. We hope that it meets the expectations of the Reviewer as it stands and will be considered worthy of publication in the Nutrients.

Best regards,

Authors.

This study compared patients with Type 1 and Type 2 diabetes undergoing hemodialysis or peritoneal dialysis for nutritional status and clinical outcomes. This research was the first to compare the results of hemodialysis or peritoneal dialysis in Type 1 and Type 2 diabetes. However, there are some issues to address.

#1 The first time an abbreviation is used in the manuscript, it should be defined. In that regard, grammatical mistakes were common throughout and should be revised. Similarly, there were a few instances where commas were used instead of periods to mark the decimal place when writing a number. Periods should be used in standard American English.

Much respected Reviewer, you are absolutely right. Thank you for pointing out the need to define abbreviations on first use and use a period before decimals. We added all necessary definitions of the abbreviations and used periods instead of commas where appropriate.

#2 Why was 22 kg/m2 used for the ideal body weight formula? Ideal body weight differs with sex and age. Since the groups are made up of various sexes and ages, does using 22 kg/m2 skew the results the GNRI?

Thank you for this remark. Assessing the nutritional status of dialysis patients is very difficult. There are also alternative ways of determining the ideal body weight. In the presented study, we used a formula using a BMI of 22 kg/m2, like other researchers [29,36,37] as presented in the methods.

#3 Since the NRI was developed for young adults, how well does it measure the nutritional status of an older adult population?

Thank you for this comment. That is true that NRI was primarily developed to identify malnutrition or the risk associated with this status and related complications in young adult surgical patients [48]. However, in a later study, NRI was used as a screening tool for malnutrition in peritoneal dialysis patients. We have supplemented the introduction section with information on the utility of GNRI and NRI in the dialysis population (lines 49-58):

“Among hemodialyzed, Geriatric Nutrition Risk Index (GNRI) has been used to assess the nutritional status as well as all-cause [4,5,28,29], cardiovascular [30], and infectious mortality [31]. GNRI has also been used to assess nutritional status and prognosis in peritoneal dialysis patients [6,32], although it has not been proven to be a sufficiently sensitive indicator for screening malnutrition in this group [33]. As a screening tool for malnutrition in PD patients, Nutrition Risk Index (NRI) can be used because it has high sensitivity but low specificity as compared to Subjective Global Assessment (SGA) [34]. Its usefulness was tested in 283 patients aged 12-65 years on peritoneal dialysis, 53 % of whom had also diabetes [34].”

 In our study, we decided to use both NRI and GNRI as well as other simple and accessible measures and indices of nutritional status showing that all of them are significantly worse among dialysis patients with type 1 DM compared to type 2 DM.

#4 Since the study only includes Caucasians, would the lack of other races limit the generalization of the results? If so, this should be addressed in the limitations section.

Much respected Reviewer, thank you for this comment. That is a very important issue. Race wasn’t in the eligibility criteria of our study. Nevertheless, all patients in the presented study were Caucasian, which reflects the high homogeneity of Polish society. Since the study population was racially homogeneous, we could apply, for example, WHR norms for the European population. However, this could limit the generalization of our results and conclusions on the global population of diabetic dialysis patients of all races. We included this comment in the limitations of the study (lines 406-408):

The last issue is that the study was conducted only in the Caucasian population, which may limit the extrapolation of conclusions to dialysis patients with diabetes of other races.”

We added in an abstract as well as in the results section that the study was conducted on Caucasian patients (lines 17 and 411).

Round 2

Reviewer 2 Report (New Reviewer)

I would like to thank the authors for their responses to my comments.  I feel like the manuscript is suitable for publication. Thank you. 

This manuscript is a resubmission of an earlier submission. The following is a list of the peer review reports and author responses from that submission.

Round 1

Reviewer 1 Report

Please find attached some comments for revision. 

Reviewer 2 Report

The study does not bring new scientific contributions to the topic. Data analysis also does not include major confounders. However, the study may be useful to support health care at the institution where it was performed (Military Institute of Medicine). Publication in a local journal is suggested.
